# OpenReview forum: "The Internal State of an LLM Knows When It's Lying"
_EMNLP/2023/Conference — EMNLP 2023 Findings_

### Official Review · Reviewer_ayRU · 2023-07-31

**Typos Grammar Style And Presentation Improvements:** see commenta bove
**Soundness:** 4

**Excitement:**

4: Strong: This paper deepens the understanding of some phenomenon or lowers the barriers to an existing research direction.

**Paper Topic And Main Contributions:**

This paper explores the hypothesis that there exists, somewhere in a large language model, a generalizable representation of the truth or falsity of a statement. The authors use a small special-purpose network to train a classifier that can convert the outputs of any hidden layer into an estimate of the truthfulness of the statement. The performance of classifiers of this type leads the authors to conclude that the generalized representation is indeed present.

In service of the hypothesis, the authors create a small corpus of examples, spanning several domains. This will be a resource for replications and future studies.

**Reasons To Accept:**

This is a very interesting and important question. The experiments are generally well-designed and informative.
The organization and narrative of the paper is excellent, making very clear what was done and why, with good examples.

**Reasons To Reject:**

There are significant numbers of small writing errors that do not impede comprehension or readability. A careful rewrite by an author who did not do much of the initial writing would fix this. These are obvious errors that would be caught by an on-line style and grammar checker.

The authors test five layers. This is a multiple compatsions setting, so statistics aiming to show that the result is real should use some kind of correction for multiple comparisons.

It is also worth, for safety, running a simple control in which the true/false labels are randomly permuted, so that each sentence is associated with the truth value of a randomly chosen counterpart. If all is well, we will find that the model is unable to predict the permuted labels with any accuracy.  This will help to establish beyond a reasonable doubt that the observed performance is due to the ability to detect truth, not to some strange effect of the behavior of large capacity models capable of learning almost anything. Possibly, doing this test is overly paranoid, but since it is cheap to do and the result is high-impact, why not?

I don't like the use of the anthropomorphic term "lying" in the title and intro. It's just about OK, because of the very good discussion of why the model might produce false statements even though both another LLM and SAPLMA can tell that they are false. But I would still prefer a more moderate phrasing in the title.

**Reproducibility:**

4: Could mostly reproduce the results, but there may be some variation because of sample variance or minor variations in their interpretation of the protocol or method.

**Reviewer Confidence:**

4: Quite sure. I tried to check the important points carefully. It's unlikely, though conceivable, that I missed something that should affect my ratings.

---

> ### Author Rebuttal · Authors · 2023-08-28
>
> We took the reviewer’s suggestion to run a test where the true/false labels were randomly permuted.  As expected, the test accuracy in this case came out to be only 50.1% (compared to test accuracy of 70.9% for SAPLMA). This is totally expected, since the model wasn't exposed to these labels during training, making its predictions purely arbitrary. However, even the accuracy on the random training data was only 62.5% (compared to training accuracy of 86.4% for SAPLMA). This indicates that our model does not have the capacity to completely over-fit the training data, and thus, must exploit structure and patterns that appear in the data. We will add a discussion of this point to the paper.
>
> Following the reviewer’s comment we will rename the title to “An LLM’s Internal State Can Indicate When False Information Is Generated” (and remove the use of the controversial word “lying”).

---

### Official Review · Reviewer_UwiC · 2023-08-03

**Typos Grammar Style And Presentation Improvements:** N/A
**Soundness:** 3

**Excitement:**

4: Strong: This paper deepens the understanding of some phenomenon or lowers the barriers to an existing research direction.

**Missing References:**

N/A

**Paper Topic And Main Contributions:**

This paper proposes a method called SAPLMA to detect whether statements generated by large language models (LLMs) like GPT are true or false. The key idea is to use the hidden layer activations of an LLM and classify whether the LLM "believes" the statement is true/false. The authors create a dataset of true/false factual statements across 6 topics, train a classifier on 5 sets and test on the hold-out set. This ensures the classifier learns to extract the LLM's internal signal of truth/falsehood rather than topic-specific knowledge.

Experiments with OPT-6.7B showed that SAPLMA achieves ~70% accuracy on held-out topics, significantly outperforming 1) few-shot prompting of the LLM itself 2) classifiers trained on BERT embeddings. The 20th hidden layer works best. SAPLMA also gets 63% accuracy on statements generated by the LLM, again outperforming baselines.

**Questions For The Authors:**

- A. Can you explain why you "expect that the BERT embeddings will perform very poorly, as they focus on the semantics of the statement rather than on whether a statement is true or false." ? It's unclear for me.
- B. Can you provide the experiment results for larger models like OPT-175B?
- C. Can you provide some ablation study or analysis to show that your classifier architecture is optimal?

**Reasons To Accept:**

- Addresses an important issue - false information generated confidently by large LLMs. Proposes a practical method to detect such false statements.
- Thorough experiments with different LLM layer activations, different topics, and human-labeled LLM-generated statements. Compares well to baselines.

**Reasons To Reject:**

- I don't know why the author claimed that "we expect that the BERT embeddings will perform very poorly, as they focus on the semantics of the statement rather than on whether a statement is true or false. " In my opinion, the classifier trained on OPT-175B and the classifier trained on BERT should be serving exactly the same function. The only difference is the model size, BERT is relatively small, while OPT-6.7B is more powerful. It's weird to say that "BERT embedding focus on the semantics, not true/falsehood". This should be your conclusion/explanation after you see the experiment results, not your claim before doing the experiment. Please explain if I misunderstand your claim here.
- As you didn't consider inputting any grounded documents when examining the truthfulness of the model, a better way is that you should only construct the examples using the pretraining data of OPT-6.7B. You showed that the final accuracy is around 70%, but we cannot know whether the remaining 30% accuracy gap is caused by #1) OPT-6.7B has not been trained on some of the new knowledge existing in your new dataset, so it's unsure (lying) about its prediction, even if the input sentence is truthful #2) the classifier is just not good enough. If you can construct a dataset in this way, we can leave out reason #1 and only focus on reason #2.
- No experiments on larger models like GPT-175B or LLaMA-65B. Unclear if findings transfer to more capable LLMs.
- The classifier architecture used seems arbitrary. No tuning of hyperparameters. More analysis is needed on optimal architectures.

**Reproducibility:**

5: Could easily reproduce the results.

**Reviewer Confidence:**

4: Quite sure. I tried to check the important points carefully. It's unlikely, though conceivable, that I missed something that should affect my ratings.

---

> ### Author Rebuttal · Authors · 2023-08-28
>
> As the reviewer points out, we do not know whether the gap in SAPLMA’s performance is due to its inability to detect if a statement is true or not, or whether statements similar to the current statements appear often in the data used to train the LLM; this is a keen observation. We show that the performance of SAPLMA varies significantly between the different topics of the dataset. This is very likely due to the distribution of the data used for training the LLM .  However, even if an exact statement appears in the dataset,the LLM is not guaranteed to predict it correctly. We note that the inventions topic is based on a Wikipedia page, which is part of the OPT training dataset; however, the performance on this topic is relatively poor. In addition, due to the nature of the true-false dataset, it is quite unlikely that any significant portion of it has changed in the last 3 years.
>
> BERT’s objective is to predict the value of masked or altered words. Therefore, its embeddings capture the semantics of the sentence. However, the veracity of a statement seems irrelevant to the task it tries to accomplish (except in very specific cases). We claim that detecting the veracity of a statement is more appropriate for generative LLMs, whose objective is to predict the next word in a sequence. We note that SAPLMA using OPT-125MB, which has fewer parameters than BERT, outperforms it by 2.4% on the true-false dataset. Nevertheless, following the reviewer’s recommendation, we will move the discussion of this hypothesis to after the results.
>
> We will provide results for a larger language model in the final version of the paper based on LLAMA 2; however, it is unlikely that we will achieve the resources required to run SAPLMA with a huge model such as OPT-175B .
>
> We do not claim that the proposed neural network architecture is optimal. In fact, we intentionally did not consider multiple architectures since we believe that the correct way to consider multiple architectures requires an additional train test validation split to avoid the winner’s curse (i.e., the selected architecture would be likely to not perform as well in practice as it did on the test set). We ran a logistic regression, which resulted in a drop of 2.5% in accuracy. We will add this to the paper.

---

### Official Review · Reviewer_DLxR · 2023-08-07

**Soundness:** 2

**Excitement:**

3: Ambivalent: It has merits (e.g., it reports state-of-the-art results, the idea is nice), but there are key weaknesses (e.g., it describes incremental work), and it can significantly benefit from another round of revision. However, I won't object to accepting it if my co-reviewers champion it.

**Paper Topic And Main Contributions:**

The paper shows evidence for the encoding of the factuality-related information in the latent representations of LMs, even when they generate nonfactual text, providing evidence for a disparity between the content of the representation space and the nature of the generated text.

**Questions For The Authors:**

* The methods section reads "We used a reliable source that included a table with several properties for each instance". What is this source?

* Did you verify the scientific-facts portion of the data, generated by ChatGPT?

* The paper reports that trying to prompt GPT directly to decide whether the statement is true of false yielded random accuracy. This is a very surprising - is that true for all categories in the data? Did you try to look at the probabilities of the actual true and false answers, rather than at the probabilities of the tokens "true" and "false"?

**Reasons To Accept:**

The paper addresses an important question -- hallucinations in LM generation and ways to identify them. The key result is nontrivial, showing that models encode the truthfulness of an answer regardless of the text they actually generate as a continuation of the prompt. The leave-on-out design of the probe is nice and is showing good generalization to different domains (although it would be nice to *also* report numbers on the same domain).

**Reasons To Reject:**

* Testing on a single dataset and a single model. At the very least, the results should be replicated on different OPT models and on different factual datasets.

* Several design decisions are not clearly motivated - see questions for the authors. Particularly, it is not clear whether the GPT-generated portion of the data was validated.

* Assuming the data is balanced, the accuracy scores reported are low. This puts into questions the claims that models "know" when the statement is true.

* It is not clear whether the information extracted by the probe is being used by the model (in the causal sense). It would be nice to include linear probes and perform linear erasure / amnesic probing experiments.

**Reproducibility:**

3: Could reproduce the results with some difficulty. The settings of parameters are underspecified or subjectively determined; the training/evaluation data are not widely available.

**Reviewer Confidence:**

3: Pretty sure, but there's a chance I missed something. Although I have a good feel for this area in general, I did not carefully check the paper's details, e.g., the math, experimental design, or novelty.

---

> ### Author Rebuttal · Authors · 2023-08-28
>
> The reviewer raises several issues and questions, which we address below.
>
> The dataset we provide is indeed completely balanced.
>
> Following the reviewer’s comment we will rename the title to “An LLM’s Internal State Can Indicate When False Information Is Generated” (and remove the use of the controversial word “knows”).
>
> Our true-false dataset is composed of 6 very different topics, and we consider sentences generated by the LLM as another dataset. The paper focuses on the OPT-6.7B model, but we also ran experiments on OPT-2.3B, OPT-1.3B, OPT-350M, and OPT-125M. We will briefly report these results. In addition, we will provide results for the LLAMA 2 model in the final version of the paper.
>
> The tables for the topics were obtained from the following sources:
> Cities: Downloaded from simplemaps.
> Inventions: Obtained from Wikipedia list of inventors.
> Chemical Elements: Downloaded from pubchem ncbi nlm nih gov periodic-table.
> Animals: Obtained from kids national geographics.
> Companies: Forbes Global 2000 List 2022: The Top 200.
>
> The scientific facts, which were obtained from ChatGPT, were verified by two human annotators. Indeed, the classification of 48 facts were questioned by at least one of the annotators. These facts were removed from the dataset. We will add this information to the paper.
>
> As another new baseline, given a statement X, we measured the probabilities (using OPT-6.7) of the sentences “It is true that X”, and “It is false that X”, and picked the higher probability (considering only the probabilities for X, not the added words). This normalizes the length/frequency factors. This results in an improvement over the zero-shot or even few-shot approach, with an accuracy of between 52.3% to 60.4%, depending on the topic, but is still way below SAPLMA’s performance. These results will be added to the paper as another baseline. We ran a logistic regression, which resulted in a drop of 2.5% in the accuracy, on average. We will add this to the paper.
>
> Regarding reproducibility of our results, we note that the code and dataset were submitted as the supplementary material and will be made public if accepted. We could not make the data public due to the anonymity period of EMNLP. Furthermore, this work was already reproduced by multiple scholars worldwide, who contacted us for obtaining the data and code.

---

### Official Review · Reviewer_72bu · 2023-08-09

**Soundness:** 3

**Excitement:**

4: Strong: This paper deepens the understanding of some phenomenon or lowers the barriers to an existing research direction.

**Missing References:**

The following paper also tackles the problem of having a LM critique on its own predictions: Language Models (Mostly) Know What They Know (https://arxiv.org/abs/2207.05221). While it mostly concerns the problem of calibration of logit likelihoods, the two papers are of similar spirits and the methods may serve as additional baselines.

**Paper Topic And Main Contributions:**

The paper describes a method to get a model to identify the veracity of its own generations at a low additional cost. It shows that with a simple classifier on top of certain layers of a model's output hidden states, the veracity prediction of model generations can be learned with a modest amount of training data, and is generalizable to unseen domains.

**Questions For The Authors:**

I have a few questions to the authors:

A. Regarding BERT classifiers and the strategy of using generation probabilities, what would the results be, if they are posed a simplified task of comparing pairs of true / false statements? (Given the construction method of the dataset, the true and false statements should be in pairs) This would serve as a measure to the information these perplexity/pseudo-perplexity values have on the veracity of statements, if the length/frequency factors etc. could be marginalized.

B. Given the difference in accuracy values with the change in thresholds, it feels it would ultimately be better to report AUC values instead of these accuracy values, for the AUC values are a more comprehensive measure of discriminative power along the spectrum of thresholds. Thoughts?

C. Considering future work, is it possible for the LM to tell at which tokens did its own outputs begin to derail?

**Reasons To Accept:**

The problem targeted by this paper, the hallucination issue and improving trust in LLMs, is one of great importance to the deployment of LLMs. The proposed method shows promising performance with lightweight architecture, experiments are thorough in the different conditions, generalization from artificial training data to actual model generations is also covered.

In the analysis of why a model would generate false claims even when they are able to discern them from true ones, the first reason discussed is also inspiring.

**Reasons To Reject:**

The method is only evaluated on a dataset that the authors created with the paper, where the exact sources of information for these datasets, as well as the construction details (e.g. how the tables are turned into sentences, etc.), are not mentioned (lines 324-325)

While the authors compared with a BERT classifier and few-shot self-correction, the pool of baselines feels a bit thin, with a lack of external baselines tackling the same problem (e.g. the missing reference listed below).

The scope of the paper is limited to simple factoid sentences, where the veracity of factoid claims embedded in more complex utterances, or the veracity of the more complex utterances themselves are not covered.

A number of presentation improvements could be made, for instance, the second reason-for-generation failure in lines 105-111 is not obvious, need further support.

**Reproducibility:**

4: Could mostly reproduce the results, but there may be some variation because of sample variance or minor variations in their interpretation of the protocol or method.

**Reviewer Confidence:**

2: Willing to defend my evaluation, but it is fairly likely that I missed some details, didn't understand some central points, or can't be sure about the novelty of the work.

**Typos Grammar Style And Presentation Improvements:**

lines 018-020: repetitive to lines 008-010;
line 118: capital M in Model;
line 205: multilingual
line 222: text
line 267: LM/LLM, not LMM

---

> ### Author Rebuttal · Authors · 2023-08-28
>
> The reviewer raises several specific questions for the authors – we address these here:
>
> A. As another new baseline, given a statement X, we measured the probabilities (using OPT-6.7) of the sentences “It is true that X”, and “It is false that X”, and picked the higher probability (considering only the probabilities for X, not the added words). This normalizes the length/frequency factors, as suggested by the reviewer. This results in an improvement over the zero-shot or even few-shot approach, with an accuracy of between 52.3% to 60.4%, depending on the topic, but is still way below SAPLMA’s performance. These results will be added to the paper as another baseline. Note that, if deployed in practice, each method will only receive a single sentence at a time. While it is possible to automatically generate two competing sentences (e.g., it is true that X, and it is false that X), it is not possible (in practice) to obtain a complementing sentence from a dataset.
>
> B. We agree that AUC captures the performance of the methods under multiple thresholds and thus we also report AUC. However, if our system were to be used in practice, one must first determine the threshold based on some data and then use that threshold for inference. We are interested to know how well such a system would perform in practice.
>
> C. The reviewer raises an interesting idea for future work. Our current attempt is to detect which sentence generated by an LLM contains false information, so that it can be marked or deleted and a new sentence generated instead. We agree that a more interesting (and challenging) task would be to detect which token caused the sentence to begin leaning toward a false statement. One can envision a system that first attempts to correct a single word, then, if it fails a few times it corrects a full sentence, and then if it fails a few times, it regenerates a paragraph, and so on. This might be somewhat similar to how humans generate text. We note that while humans many times delete and rewrite sentences, this data is usually not recorded, and thus, current LLMs never predict a ‘delete’ token.
>
> We thank the reviewer for the pointer to relevant work, which will be added to the paper.
>
> Regarding reproducibility of our results, we note that the code and dataset were submitted as the supplementary material and will be made public if accepted. We could not make the data public due to the anonymity period of EMNLP. Furthermore, this work was already reproduced by multiple scholars worldwide, who contacted us to obtain the data and code, which we happily provided.

---

### Meta-Review · Area_Chair_YnL4 · 2023-09-15

**Recommendation:** 4

**Metareview:**

This paper received slightly disparate scores: 3,2,3,4 (soundness) and 4,3,4,4 (excitement). Three of the reviewers have rated the soundness & excitement as Good or Strong, whereas one reviewer (R2 / DLxR) raises objections and provides lower scores.

Condensing the reviews, the following major strengths and weaknesses were noted:

Strengths:
- everyone agreed that the paper targets a key issue in LLMs
- the reviewers specifically praised various aspects of the paper, such that nontriviality of the result (R2), thorough experiments (R3), and good presentation (R4)

Weaknesses:

- testing on a single dataset (R1,R2) and a single model (R2,R3), with some concerns about lacking details on the newly constructed dataset (R1)
- a limited set of baselines (R1)
- scope is limited to simple factoid sentences (R1)
- a range of unclear design decisions were mentioned (R2,R3)
- concerns about the link between extractability with a probe and causality on the model behavior (R2)

R2 provided a low soundness score. However, the concerns they mentioned were overall similar to those mentioned by the other reviewers, and the authors addressed them to some extent. I thus believe that these soundness concerns do not warrant rejecting the paper.

---

### Decision · Program_Chairs · 2023-10-07

**Decision:**

Accept-Findings

**Comment:**

This paper received slightly disparate scores: 3,2,3,4 (soundness) and 4,3,4,4 (excitement). Three of the reviewers have rated the soundness & excitement as Good or Strong, whereas one reviewer (R2 / DLxR) raises objections and provides lower scores.

Condensing the reviews, the following major strengths and weaknesses were noted:

Strengths:
- everyone agreed that the paper targets a key issue in LLMs
- the reviewers specifically praised various aspects of the paper, such that nontriviality of the result (R2), thorough experiments (R3), and good presentation (R4)

Weaknesses:

- testing on a single dataset (R1,R2) and a single model (R2,R3), with some concerns about lacking details on the newly constructed dataset (R1)
- a limited set of baselines (R1)
- scope is limited to simple factoid sentences (R1)
- a range of unclear design decisions were mentioned (R2,R3)
- concerns about the link between extractability with a probe and causality on the model behavior (R2)

R2 provided a low soundness score. However, the concerns they mentioned were overall similar to those mentioned by the other reviewers, and the authors addressed them to some extent. I thus believe that these soundness concerns do not warrant rejecting the paper.